# Vaccine-Related Autoimmune Hepatitis: Emerging Association with SARS-CoV-2 Vaccination or Coincidence?

**DOI:** 10.3390/vaccines10122073

**Published:** 2022-12-04

**Authors:** Ioannis P. Trontzas, Konstantinos G. Kyriakoulis, Ioannis A. Vathiotis, Alexandros Syrigos, Georgios Kounadis, Sofia Siasiakou, Garyphalia Poulakou

**Affiliations:** 13rd Department of Internal Medicine, ‘SOTIRIA’ General Hospital of Chest Diseases, National and Kapodistrian University of Athens, 11527 Athens, Greece; 2Department of Pathology, Yale University School of Medicine, New Haven, CT 06519, USA

**Keywords:** SARS-CoV-2, COVID-19, vaccine, vaccination, hepatitis, autoimmune hepatitis, AIH, autoimmune liver injury, immune-mediated liver injury, review

## Abstract

Background: There is an increasing number of liver injury cases resembling autoimmune hepatitis (AIH) following SARS-CoV-2 vaccination; however, an association has not yet been established. Methods/Materials: A literature review was performed to identify articles regarding the association of AIH with vaccination, emphasizing on SARS-CoV-2 vaccines, and the proposed mechanisms. We then performed a literature search for AIH-like cases following SARS-CoV-2 vaccination, and we evaluated the included cases for AIH diagnosis using simplified diagnostic criteria (SDC), and for vaccination causality using the Naranjo score for adverse drug reactions. Results: We identified 51 AIH-like cases following SARS-CoV-2 vaccination. Forty cases (80%) were characterized as “probable”, “at least probable”, or “definite” for AIH diagnosis according to SDC. Forty cases (78.4%) were characterized as “probable”, four (7.8%) as “possible”, and three (5.8%) as “definite” for vaccine-related AIH according to the Naranjo score. Conclusion: SARS-CoV-2 vaccine-related AIH carries several phenotypes and, although most cases resolve, immunosuppressive therapy seems to be necessary. Early diagnosis is mandatory and should be considered in any patient with acute or chronic hepatitis after SARS-CoV-2 vaccination, especially in those with pre-existing liver disease.

## 1. Introduction

The coronavirus disease 2019 (COVID-19) pandemic prompted an international effort for swift vaccine delivery. The Food and Drug Administration (FDA) granted emergency use authorization for the first vaccine in December 2020 [1], one year after the identification of the first COVID-19 case in Wuhan, China [2]. Currently, several COVID-19 vaccines are available [3]. Due to the rapid marketing of these agents, the focus has turned into continuous safety surveillance. About 68% of the global population has received at least one vaccine dose [4] and monitoring authorities have not issued any significant safety warnings so far [5]. Some rare adverse events (AEs) have been recognized, including post-vaccination anaphylaxis, myocarditis and pericarditis, thrombosis with thrombocytopenia syndrome, and Guillain–Barre syndrome [6].

Besides the above-mentioned AEs, there are many other events reported as possible adverse drug reactions (ADRs). In the UK, a total of 172,901 Yellow Cards for the COVID-19 vaccines were reported and reviewed as of 24 August 2022 [7]. Moreover, a simple search in the PubMed database with the algorithm “COVID-19 vaccination adverse events” will bring up more than 14,000 results as of September 2022.

The reporting of AEs is an integral part of drug safety monitoring. A careful case-by-case examination should be performed to protect public health and ensure compliance with vaccination.

In line with this surveillance approach, our interest has focused on the emerging reports for autoimmune hepatitis (AIH) following SARS-CoV-2 vaccination. AIH is a relatively rare liver disease, and the diagnosis is based on clinicopathologic characteristics and specific diagnostic criteria [8]. AIH may develop after the administration of many drugs (drug-induced AIH (DI-AIH)). DI-AIH is a complex condition with a broad disease spectrum [9]. Vaccination-induced AIH has been reported in the past [10,11,12,13,14,15,16]; however, mass vaccination against SARS-CoV-2 has brought up an increasing number of new reported cases.

Regarding post-vaccination hepatitis, some important questions are: (i) Is there a direct, even rare, causal association of AIH with SARS-CoV-2 vaccines? (ii) Is this association attributed to a specific type of vaccine? (iii) If there is a causal relationship between the two, does this represent “true” AIH requiring long-term immunosuppressive management or a transient hepatitis with autoimmune features? (iv) What are the risk factors and what should the recommendations be?

In this narrative literature review, we sought to address the above-mentioned questions and tried to provide the available evidence for their answers. We revised the knowledge regarding the association of AIH with vaccination, we reviewed the proposed mechanisms, we conducted a literature review for cases of AIH following SARS-CoV-2 vaccination, we assessed the causal relationship between the two, and finally, we aimed to alert for the management and ongoing monitoring of similar cases.

## 2. Autoimmune Hepatitis and Its Association with Vaccination

AIH is an acute or frequently chronic liver disease of unknown etiology and is characterized by hypergammaglobulinemia (polyclonal, particularly immunoglobulin G (IgG)), autoantibodies, interface chronic and progressive hepatitis, and a favorable response to immunosuppressive treatment [8,17]. It should be distinguished from the more common viral, non-alcoholic fatty liver/steatohepatitis, drug-induced, and alcohol-related forms of hepatitis [8]. The annual incidence and prevalence are estimated at 0.9–2 cases/100,000 and 11–25/100,000, respectively, with a female predominance (3.6:1) [18,19,20]. Its clinical manifestations are variable, not specific, and may range from asymptomatic to fatigue, malaise, icterus, abdominal pain, or even fulminant hepatitis [8,21]. AIH may present concurrently with other autoimmune diseases (ADs), and their symptoms may (erroneously) be appreciated as extrahepatic manifestations in many cases [8,22]. Type 1 AIH is typically associated with the presence of antinuclear antibodies (ANAs), anti-smooth muscles (ASMAs), and anti-soluble liver antigens (SLAs), while type 2 is associated with the presence of liver/kidney microsomes (LKMs) and liver cytosol antibody type 1 (LC1) [8,17,21]. The diagnosis is challenging and is based on clinicopathologic characteristics. The presence of hepatitis is a requisite for an AIH diagnosis as implied by The International Autoimmune Hepatitis Group’s revised diagnostic criteria (RDC) for AIH and simplified diagnostic criteria (SDC) [23,24]. The histological hallmarks include interface hepatitis, portal lymphoplasmacytic infiltrates, rosette formation, perivascular necrosis, and moderate to severe necro-inflammatory lesions in the lobules [17,25,26]. It should be noted, though, that these findings are not pathognomonic. Management requires long-term immunomodulation [8,21,22,26]. It is thus clear that idiopathic AIH is a specific clinical entity and should be distinguished from immune mediated drug-induced liver injury (IM-DILI) or other forms of hepatitis with autoimmune features, as will be discussed in the next section [27].

There are several reports in the literature implicating vaccination in AIH development. This association has been described with vaccines against various agents, including influenza [12,13]; the hepatitis A virus (HAV) [11,14,15,16]; the hepatitis B virus (HBV) [15,16]; the human papilloma virus (HPV) [10]; yellow fever [11]; and diphtheria, pertussis, and tetanus (DPT) [11,15].

In most of these reports, the clinical presentation is similar, with signs of acute hepatitis. Jaundice is the most frequent initial symptom. The expression of AIH-related autoantibodies and the increase in IgG is variable. The histological findings on liver biopsy are compatible with or typical for AIH. In most cases, long-term immunosuppressive treatment is necessary for remission, indicating an idiopathic AIH phenotype, rather than transient hepatitis with autoimmune features [10,11,12,13,14,15,16]. Interestingly, besides influenza and HPV vaccines, HAV vaccination is involved in all other cases either as a single-agent vaccine or as part of a multiple vaccination, suggesting a possible etiological correlation of HAV with AIH. Finally, it must be noted that all the reported cases refer to viral vector vaccines, either live-attenuated or inactivated [10,11,12,13,14,15,16]. This reference may be opposed to the suggestion that liver autoimmunity (among other autoimmune phenomena) may be triggered solely by mRNA vaccines.

In addition, there are reports associating AIH with other types of vaccines [28], including measles–mumps–rubella, polio, and typhoid; however, in our literature search, we failed to retrieve the original sources regarding these cases.

The mass vaccination against COVID-19 has brought up several cases of post-vaccination AIH. Rigorous literature reporting of these cases has been reflected in the publication of several systematic reviews on the topic. Chow et al. recently published a systematic review describing 32 cases of AIH-like syndromes following SARS-CoV-2 vaccination [28]. Another systematic review by Roy et al. described 21 cases of post-SARS-CoV-2 vaccination immune-mediated liver injury [29]. A more recent systematic review by Alhumaid et al. recognized 138 cases of AIH, including four cases with pre-existing AIH [30]. Moreover, according to the WHO Program for International Drug Monitoring (PIDM), 10,388 cases of hepatobiliary ADRs were reported following SARS-CoV-2 vaccination, of which 646 (6.2%) were AIH [31].

## 3. Proposed Mechanisms of Autoimmune Hepatitis following SARS-CoV-2 Vaccination

With the increasing reports of SARS-CoV-2 vaccine-related AIH, comprehension of the pathophysiological mechanisms may help in the characterization of this condition. The exact interplay between vaccines, the immune system, and hepatocytes remains unknown so far. As a result, the proposed mechanisms are indirect conclusions based on the knowledge of idiopathic AIH, on immune pathways described in other SARS-CoV-2 vaccine-related AEs, on autoimmune complications of natural COVID-19 infections, and on the results of small studies or isolated case reports.

The variability across clinical presentations and the wide range of latency times between vaccine administration and symptom onset, along with the variable number of doses administered before the liver injury diagnosis, suggest that there is an interplay of different mechanisms that may contribute to AIH. A series of interconnecting autoimmune events after vaccination may eventually lead to AIH (Figure 1).

### 3.1. Molecular Mimicry

In idiopathic AIH, it has been proposed that molecular mimicry may play an important role. The mechanisms of molecular mimicry may become activated in genetically susceptible individuals upon an immune system encounter with antigens that are structurally similar to self-proteins [8,32]. A genetic predisposition for AIH has been associated with the class II HLA phenotype, and more specifically with the DRB1 gene locus [33]. The hypothesis of molecular mimicry is better incorporated in type 2 AIH, where the main target of anti-LKM1 antibodies is the liver enzyme cytochrome P450 2D6 (CYP2D6). Some CYP2D6 DNA sequences exhibit a high resemblance to the protein by-products of HCV and of the Herpesviridae family [8,34]. In fact, anti-HCV antibodies may be found in 50% of patients with type 2 AIH [35,36]. Autoimmune hepatocyte destruction involves the erroneous recognition of self-antigens by antigen-presenting cells, many of which reside in the liver, such as sinusoidal endothelial cells, Kupffer cells, and dendritic cells. As such, the presentation of a target antigen to helper and cytotoxic T-cells may take place within the liver, sparing the secondary lymphoid organs from presentation and promoting immune tolerance [37,38].

The mechanisms of molecular mimicry may play a role in post-SARS-CoV-2 vaccination AIH, as antigen cross-reactivity has already been described in natural COVID-19 infections [39]. Severe cases of COVID-19 are characterized by autoinflammatory dysregulation with consequent tissue damage, and it appears that the viral spike protein plays a role in this [40,41]. Moreover, it has been recognized that there is a shared heptapeptide between the human proteome and the viral spike glycoprotein [39,40]. It is thus reasonable to assume that spike-directed antibodies induced by vaccination may recognize cross-reacting self-antigens or shared antigenic protein sequences, and this molecular mimicry may trigger autoimmune conditions in predisposed individuals.

### 3.2. Autoimmune/Inflammatory Syndrome Caused by Adjuvants

Autoimmune/inflammatory syndrome caused by adjuvants (ASIA) has been described by Shoenfeld and Agmon-Levin as a possible mechanism for vaccine-induced autoimmunity [42]. Vaccine adjuvants may direct autoimmune processes through antigen protection and translocation to the lymph nodes, recognition by Toll-like receptors (TLRs), and activation of cellular immunity [43]. Viral vector SARS-CoV-2 vaccines contain ingredients such as L-histidine, L-histidine hydrochloride monohydrate, magnesium chloride hexahydrate, polysorbate 80, ethanol, sucrose, sodium chloride, disodium edetate dihydrate (EDTA), and water for injection, while mRNA vaccines use liposomal nanoparticles [44,45]. Those adjuvants may lead to ASIA; however, more detailed studies into SARS-CoV-2 vaccines are needed to establish adjuvant causality.

### 3.3. Direct mRNA Effects

In addition to adjuvants embedded in mRNA vaccines, mRNA itself triggers a series of steps in the process of protein translation, and each of these steps may lead to the activation of autoimmune processes in a similar manner as in ASIA. One major event in the process of protein translation is the binding of mRNA to pattern recognition receptors (PRRs) [44]. Several PRRs, such as TLRs, retinoic acid-inducible gene-I (RIG-I), and melanoma differentiation-associated protein 5 (MDA5), enable the recognition of RNA chains found in the cytosol or in endosomes. The final outcome of this process is the activation of several pro-inflammatory cascades, including the assembly of inflammasome, the type I interferon (IFN) response, and the nuclear translocation of the nuclear factor (NF)-Kb, which are known to induce autoimmunity [44]. 

### 3.4. Bystander Hepatitis and Autoreactive Lymphocytes

AIH has been described in natural COVID-19 infections [46], among other autoimmune phenomena [40]. Some of these cases have been attributed to bystander hepatitis triggered by the activation of systemic cytokines/chemokines, which also have critical roles in the inflammatory process of AIH, such as TNF-α, IFN-γ, IL-1β, IL-6, and IL-10, and the subsequent activation of autoreactive B- and T-cells [47]. The release of these cytokines may also take place after SARS-CoV-2 vaccination, leading to autoimmune liver injury. Vaccine-induced spike-specific CD8 T-cells were found in peripheral blood along with extensive CD8 liver infiltration in a case of post-SARS-CoV-2 vaccine AIH, enhancing the suspicion that autoreactive lymphocytes may contribute to the pathogenesis of AIH [48].

### 3.5. Epitope Spreading Hypothesis

An alternative or contributing explanation could be the epitope spreading (ES) hypothesis. ES has been associated with vaccination in the past and may contribute to SARS-CoV-2 vaccine-related AIH. In fact, ES has been employed in vaccine manufacturing (e.g., *Haemophilus influenzae B* polysaccharide vaccines). ES is the development of an immune response to epitopes distinct from, and non-cross-reactive with, the dominant epitope, and it is rather a protective feature of the immune system [49]. In autoimmunity, ES works through the diversification of epitopes recognized by the immune system, and it may appear to both B- and T-cells. Endocytic processing, antigen presentation, and somatic hypermutation drive ES and preserve the immune response in autoimmune processes [50].

### 3.6. Overt Autoimmunity Triggered by Vaccination

It is also worth mentioning that early autoimmune responses after vaccination may reflect the presence of overt autoimmunity, which is triggered by vaccination, rather than autoimmunity caused de novo by vaccines [44,51]. In many post-SARS-CoV-2 vaccine-related AIH cases, the elevation of IgG and the detection of autoantibodies were documented soon after vaccination [48,52,53,54,55,56,57,58,59,60,61,62,63,64,65,66,67,68,69,70,71,72,73,74,75,76,77,78,79,80,81,82,83,84,85,86,87,88], which warrants possible risk factor screening on selected individuals before vaccination.

## 4. Reported Cases of Autoimmune Hepatitis following SARS-CoV-2 Vaccination

To investigate the extent of increasing reports of AIH following SARS-CoV-2 vaccination, we performed a literature search in the PubMed database from August to October 2022, using the following algorithm: (“autoimmune hepatitis” OR “autoimmune hepatitis-like” OR “immune-mediated hepatitis” OR “hepatitis” OR “hepatitis-like”) AND (“COVID-19 vaccination” OR “COVID-19 vaccine” OR “SARS-CoV-2 vaccination” OR “anti-SARS-CoV-2 vaccination” OR “following COVID-19 vaccination” OR “post-COVID-19 vaccination”). We included peer-reviewed articles in the English language. After screening by four authors who worked independently (IPT, KGK, IAV, and AS) and consensus with three senior authors (GK, SS, and GP), we excluded articles referring to post-vaccination liver injury without sufficient evidence for the diagnosis of hepatitis with autoimmune features (negative autoantibodies, no elevation of IgG, not typical biopsy). Further articles were added through the “snowball procedure” from the reference lists of the published systematic reviews [28,29,30].

Overall, 51 reports of AIH following SARS-CoV-2 vaccination were included [48,52,53,54,55,56,57,58,59,60,61,62,63,64,65,66,67,68,69,70,71,72,73,74,75,76,77,78,79,80,81,82,83,84,85,86,87,88]. The main characteristics of the cases are illustrated in Table 1.

### 4.1. Demographic Data and Concurrent Diseases

The median age was 57 years (interquartile range (IQR): 21–82) with female predominance (74.5%). Among those patients, 17 (33.3%) had a history of AD, pre-existing AIH, other liver diseases, or the use of immunostimulatory treatment. Besides pre-existing AIH, these conditions included primary sclerosing cholangitis (PSC), primary biliary cholangitis (PBC), chronic HCV infection, liver cirrhosis, liver transplantation (due to AIH), Hashimoto thyroiditis, sarcoidosis, ulcerative colitis, family history of AD, immunotherapy for cancer, and one patient had history of jaundice episodes of spontaneous resolution. 

### 4.2. Vaccination History

Most patients (70.6%) were vaccinated with an mRNA agent, while one patient had completed his first vaccine sequence with an inactivated agent and his symptoms presented after the third dose, which was administered with an mRNA vaccine [57]. Nineteen patients (44.2%) presented with symptoms after the first vaccine shot, 15 (39%) after the second, and two (4.6%) after the third dose. Worthy of mention is that in seven cases (16.2%), the symptoms presented after the first dose and flared after the second. 

### 4.3. Clinical Presentation

The most common symptoms upon presentation were jaundice (57%), fatigue (27.5%), nausea and/or vomiting (25.5%), dark urine (21.6%), pruritus (17.7%), anorexia (15.7%), acholic stools (9.8%), abdominal pain (11.8%), and fever (9.8%). Additionally, in 25% of the cases, the initial presentation included other symptoms, such as diarrhea, headache, weight loss, leg edema, arthralgia, and disorientation. Two patients eventually developed liver failure and one of them disseminated intravascular coagulation (DIC). The symptoms presented after a median latency period of 10 days (1–53) after vaccination. 

### 4.4. Laboratory Investigation

On an initial liver function test (LFT) evaluation, 41 patients (89.1%) demonstrated a mixed pattern of liver injury, with abnormalities in both aminotransferases and cholestatic enzymes, and in four cases (8.7%), the patients demonstrated only an aminotransferase increase. Out of 51 patients, autoantibodies were detected in 43 (84.3%), including ANA (76.5%), ASMA (23.5%), and other (antimitochondrial, anti-SSA, anti-gastric parietal, anti-dsDNA, anti-actin, anti-neutrophilic cytoplasmic, or anti-LC1) (25.5%). IgG was borderline elevated or higher in 37 cases (72.5%). In five cases (10%), human leukocyte antigen (HLA) phenotyping was performed, and the gene loci detected included A1, B8, DRB1, DQA1, DQB1, DR3, and DR4. Flow cytometry of the peripheral polymorphonuclears revealed circulating anti-spike CD8 T-cells in one case [48]. 

### 4.5. Liver Histology

Fifty patients had available liver biopsies, out of which 47 (92.2%) had histopathological findings compatible with or typical of AIH. In nine cases (17.6%), eosinophilic infiltrates were found on liver tissue; in another case, the biopsy was more consistent with DILI rather than AIH; and in one case, there were findings of an AIH/PBC variant. Furthermore, a mass cytometry of the liver tissue was performed in one case, which revealed panlobular infiltration with CD8 T-cells [48]. 

### 4.6. Management and Outcomes

Forty-two patients (82.3%) were treated with corticosteroids, and in 13 cases (25.5%), azathioprine was added. Other regimens used were plasmapheresis and ursodeoxycholic acid. Four patients did not receive immunomodulatory treatment and the hepatitis resolved spontaneously. Overall, all patients responded well except for four patients (8%), who passed due to fulminant hepatitis and liver failure (one developed DIC and one developed sepsis).

### 4.7. Evidence from Excluded Studies and Reports

In comparison with the published systematic reviews [28,29,30], we have excluded several retrospective studies and case reports from our review and descriptive analysis. First, a retrospective multicenter study by Efe et al. was not included, as (i) it provided aggregate data for patients with liver injury (not specific descriptive data for patients with immune-mediated hepatitis), (ii) not all patients received a liver biopsy, and (iii) they defined autoimmune liver injury solely according to autoantibody status and IgG levels [89]. In this study, 45 patients demonstrated an immune-mediated phenotype of liver injury (positive autoantibodies and elevated IgG) with a median age of 49 years (30–76) and female predominance (64.4%). Four patients (8.9%) had pre-existing liver disease and fifteen (33.3%) had a history of AD. The general characteristics were not significantly different between patients with and without features of immune-mediated hepatitis. The need for corticosteroid therapy was more frequent in patients with immune-mediated hepatitis than those without (71.1% vs. 38.2%), but the median time to biochemical resolution was similar [89]. Second, a multicenter case series including 16 patients with post-SARS-CoV-2 vaccination liver injury by Schroff et al. was not included in our review due to a lack of efficient evidence for AIH-like syndrome characterization [90]. Importantly, in this retrospective series, four patients with pre-existing AIH were reported. All of them were in remission prior to vaccination [90]. In addition, a multicenter retrospective study of patients with an AIH diagnosis after SARS-CoV-2 vaccination by Rigamonti et al. was excluded, as it was not peer reviewed (poster publication) [91]. In this study, 12 patients were reported with a median age of 62 years (32–80) and without gender disparity (1:1 male to female). Six of those (50%) had a history of pre-existing AD, and nine (75%) had received an mRNA-based vaccine. An important finding of this study was that three months after treatment, only 58% had achieved a complete biochemical response to immunosuppressive treatment [91]. Finally, a case report by Nyein et al. was excluded, as it could not be characterized as an immune-mediated liver injury [92]. Interestingly, this was the first case with the detection of anti-SARS-CoV-2 specific antibodies scattered throughout liver parenchyma [92].

## 5. Causal Association or Coincidence?

After the literature review, our intention was (i) to objectively evaluate if the reviewed cases reflected AIH, and (ii) if there is causal relationship with SARS-CoV-2 vaccination. To do so, two scoring scales were exploited for the objective assessment of diagnosis and causality, which were applied to each reviewed case by four investigators (IPT, KGK, IAV, and AS), who worked independently.

### 5.1. Simplified Diagnostic Criteria for Autoimmune Hepatitis Diagnosis

The possibility of AIH diagnosis was assessed using the SDC, which is a validated diagnostic tool. We chose the SDC over the RDC due to its higher specificity (90% vs. 73%) and accuracy (92% vs. 82%) in the diagnosis of AIH, and its high negative predictive value in exclusion of other liver diseases with concurrent immune manifestations (83% vs. 64%) [93]. Moreover, the RDC requires a pre- and post-treatment evaluation, which in most cases was not feasible to calculate. The SDC scores provided in the original publications were adopted in our analysis. In all other cases, we calculated the scores based on the available data. The SDC scores for the reviewed cases are illustrated in Table 2. A case-by-case calculation of the SDC scores is provided in Appendix A. Overall, 25 cases (50%) were assessed as “definite” for an AIH diagnosis (score ≥ 7), 11 cases (25%) as “probable” (score ≥ 6), and 10 (23%) were assessed as “non-probable” for AIH (score ≤ 6). One case did not provide sufficient evidence for an SDC score calculation. [52] Collectively, 40 cases (80%) were characterized as “probable”, “at least probable”, or “definite” for an AIH diagnosis according to the SDC.

### 5.2. Naranjo Score for Adverse Drug Reactions for Vaccination Causality

For the causality investigation, we considered several ADR scores, as there are numerous such algorithms available (Jones’, Yale, Karch, Begaud, ADRAC, WHO–UMC, Bradford–Hill, etc.) [94]. The Roussel Uclaf causality assessment method (RUCAM) is a validated, frequently used tool for assessing causality in DILI [95]. Though the RUCAM is the best tool for DILI evaluation, it is rather complex, and, in many cases, the authors did not provide sufficient data for its calculation (e.g., no laboratory upper normal limits provided, no available initial and repeated liver enzyme values in many cases). The Naranjo score, another ADR algorithm, was developed to standardize the assessment of causality for all ADRs, was designed for use in controlled trials and registration studies of new medications, and its validity compared to clinical judgement has been established [94,96]. We chose the Naranjo score as it was feasible in our cases. We have to underline, though, that Naranjo finds better applicability to clinical trials than case studies, and it has a lower sensitivity (54%) and a poor negative predictive value (29%) in assessing DILI compared to the RUCAM [97].

The case-by-case Naranjo score calculation details are provided in Appendix A. The Naranjo score outcomes and the probability of ADR are illustrated in Table 2. Out of the 51 cases, 40 (78.4%) were characterized as “probable” (scores 5–8), 4 (7.8%) as “possible” (scores 1–4), and 3 (5.8%) as “definite” (scores ≥ 9). In four cases (7.8%), we did not have sufficient evidence to accurately estimate the probability [52]; thus, they were characterized as “possible/probable” based on their potential minimum and maximum scores. 

### 5.3. Definite Cases of Autoimmune Hepatitis Associated with SARS-CoV-2 Vaccination

Two cases that were characterized as “definite” ADR by Naranjo were also “definite” by the SDC for an AIH diagnosis [48,80]. The third “definite” case by Naranjo was characterized as “at least probable” for AIH in the SDC [82]. However, the presentation of hepatitis with immune features, both after the first and second vaccine dose, and the prompt improvement after corticosteroid administration, significantly raised suspicions for a vaccine-induced autoimmune liver injury in the latter case [82]. In all three cases, the common characteristics included symptom manifestation after each dose, rapid improvement with corticosteroids, a positive serology for AIH-related autoantibodies, and a liver histology typical for AIH. These three cases may reflect the low sensitivity of Naranjo, as it seems that the addition of flared symptoms after the second dose bypassed the inherit Naranjo limitations on assessing DILI. Besides the mechanistic issues of the scores used, the typical presentation of AIH in these three cases and the direct correlation with vaccination consolidates the hypothesis that autoimmune vaccine-related phenomena may lead to an AIH-like syndrome.

## 6. Variable Phenotypes of Autoimmune Hepatitis following SARS-CoV-2 Vaccination

DI-AIH may present with diverse clinical characteristics, laboratory findings, and histological findings. Thus, it is often difficult to distinguish between vaccine-related AIH and DILI.

In fact, there are several possible combinations of DILI and AIH, with at least three clinical scenarios as proposed by Weiler-Normann and Schramm [27,98]; (i) AIH with DILI, where the reactivation of a known AIH occurs upon the introduction of a new drug; (ii) DI-AIH, where a patient without a previous diagnosis presents with persistent AIH after DILI and the need for long-term immunosuppression; and (iii) IM-DILI, where acute or chronic liver injury with autoimmune features occurs and may resolve or become quiescent with drug withdrawal [99]. In the cases where IM-DILI persists after drug withdrawal, the administration of immunosuppressants is necessary and the condition is indistinguishable from DI-AIH [9,98]. The differentiation of DI-AIH and IM-DILI is in most cases difficult, as the clinical presentation may be identical. Little evidence may help with the diagnosis, such as the presence of eosinophilic infiltrations in liver biopsy, the normal values of IgG, or the absence of AIH-related autoantibodies in the case of IM-DILI [9,100]. In addition, a mixed autoimmune type of hepatitis with sharing features of DI-AIH and IM-DILI may be recognized. This mixed type has been described in cases of minocycline-induced hepatitis, where the patients demonstrate a complete response after immunosuppression; however, they experience chronic hepatitis after corticosteroid withdrawal [101,102]. It is thus clear that although these mechanisms are distinct, they may also be interconnected [100]. 

In the case of SARS-CoV-2 vaccines, it is in our belief that all the described mechanisms may be involved in liver injury, and the crucial clinical questions are (i) which patients will need immunosuppression, (ii) what the risk for transition to chronic hepatitis is, (iii) which of these patients will be rechallenged with vaccination, and (iv) what we can do to screen this population.

### 6.1. Immune-Mediated Drug-Induced Liver Injury following SARS-CoV-2 Vaccination

There are several reports resembling IM-DILI in patients developing clinical hepatitis following SARS-CoV-2 vaccination, without any autoimmune feature (such as the presence of autoantibodies, an IgG increase, a compatible liver histology, or a good response to immunosuppressants) or with a varying expression of these features [54,62,79,89,90,92,103]. These cases, though severe at times, are usually self-resolving without further management needed. Despite the generally mild phenotype, this condition lies still in a field of uncertainty, so the close monitoring of LFTs is essential until hepatitis resolution.

### 6.2. Flare of Pre-Existing Autoimmune Hepatitis following SARS-CoV-2 Vaccination

Another recognized entity after SARS-CoV-2 vaccination is AIH with DILI [52,56,85,87,90]. Those patients are usually easily identified, as they are closely followed for their underlying condition. It is crucial to immediately manage them with immunosuppression until remission. Rechallenging with the SARS-CoV-2 vaccine after a vaccine-related AIH flare is a difficult decision and must be individualized. Depending on the intensity of the flare, vaccine rotation along with close monitoring may be reasonable.

### 6.3. New-Onset Autoimmune Hepatitis following SARS-CoV-2 Vaccination

The majority of the recognized cases resemble DI-AIH [48,55,57,58,60,65,66,67,68,69,70,72,76,80,81,85,86,88]. SARS-CoV-2 vaccination appears to associate with a de novo AIH-like syndrome in many cases; however, differences with idiopathic AIH may be observed. The management of these patients usually requires immunosuppression. Despite that, remission is achieved soon and flares upon corticosteroid withdrawal are not observed. Although this condition does not seem to become chronic, a longer surveillance period is required to better characterize it. 

### 6.4. Mixed Autoimmune Type of Hepatitis following SARS-CoV-2 Vaccination

Finally, a mixed type of de novo post-SARS-CoV-2 AIH with sharing features of DI-AIH and IM-DILI may be also recognized. This condition is reflected on the diverse findings of autoantibody expression, IgG levels, and histological examination seen in many reports. The mixed pattern of liver injury is better recognized in histological findings, where there are overlapping phenotypes of AIH (interface hepatitis, rosette formation) and DILI (eosinophilic infiltrations, predominance of CD8 T-cells, varying degrees of plasmacytic infiltrations) [48,61,62,64,71,73,75,77,78,81,82,84]. Immunosuppressive treatment appears to be necessary in this case as well. Vaccine re-introduction approach remains uncertain. We believe that individualized decisions and close monitoring is the most reasonable approach.

## 7. Discussion

Autoimmune phenomena following SARS-CoV-2 vaccination have been rarely recognized [6]. COVID-19 infection has been linked with autoimmune complications as well [40]. Increasing reports of AIH follow a SARS-CoV-2 vaccination alert for another emerging autoimmune AE. 

Idiopathic AIH accounts for 1–2/100,000 new cases annually [18,19,20]. The application of these numbers to vaccinated individuals for COVID-19 would result in many patients that would be expected to develop AIH, whether vaccinated or not. As so, causality is difficult to prove, and vigorous post-marketing surveillance is needed. 

In this narrative literature review, we tried to assess the extent of this increasingly reported phenomenon. The main limitation of our review is the lack of a systematic approach. However, a non-systematic search enabled our team to comprehensively evaluate not only the extent of the reports, but also the true representation of hepatitis with autoimmune features among these cases. Moreover, we tried to calculate the SDC scores to identify which cases qualified for an AIH diagnosis. We then tried to assess causality using a validated scale for ADRs. Here lies another limitation, as the RUCAM scale, a validated tool for DILI assessment, was not applicable to our cases. As a result, we chose the Naranjo score for ADRs, an algorithm that is less sensitive compared to the RUCAM. While our results from the SDC were strongly conclusive for a diagnosis of AIH (80% “probable” or “definite” diagnosis), inconclusive results on causality may reflect the low sensitivity of Naranjo for the characterization of DILI [97]. 

Although the current literature (including our review) fails to provide sufficient evidence for the association (rather than causality) of SARS-CoV-2 vaccination with AIH, the number of emerging reports is at least notable, especially compared with published cases of AIH related with other vaccines. In our review, we have included 51 cases, having applied rather strict criteria, and excluded some retrospective studies and case series, whereas the reports associating AIH with influenza, HAV, or other vaccines are significantly fewer. Even greater numbers are reported in vigilant databases; however, it is difficult to assess the exact diagnosis and causality of these cases [6,31]. Another factor suggesting a possible association is the short latency time (10 days) for symptom onset after vaccination. As a result, the possibility of an emerging autoimmune phenomenon should be carefully appreciated. As the condition is rare, retrospective studies, case series, and isolated reports provide much of the evidence, and their reporting can help health care providers understand the varied presentations and management approaches. Thus, we encourage the reporting of similar cases, and we propose the use of validated tools, such as the SDC for AIH diagnosis and the RUCAM for DILI causality, which will yield objectivity to future reports.

It becomes evident that disease presentation is rather diverse. Out of the 51 reports, there are patients with pre-existing AIH who experienced a flare after vaccination, others with findings typical for de novo AIH, and cases more consistent with an IM-DILI with varying degrees of autoimmune features. This diverse presentation may resemble a complex interplay of interconnecting pathophysiological mechanisms that lead to autoimmune liver injury, including molecular mimicry, ASIA, bystander hepatitis, ES, and the activation of innate immunity [39,40,41,44,48,50]. Furthermore, liver injury resolves in most cases with or without immunosuppression. These findings, which are consistent with other literature reviews [28,29,30], are in contrast with the idiopathic AIH phenotype, which is a chronic condition requiring long-term immunosuppression. It is thus reasonable to assume that we face a new entity of IM-DILI presenting with a wide range of autoimmune characteristics, rather than idiopathic AIH. Additionally, this diverse presentation may account for disease under-reporting, as milder cases may be asymptomatic and “masked”.

Our efforts should be focused on better characterizing this new entity out of the increasing literature evidence. Some of the common characteristics between our and other literature reviews is a female predominance, the correlation with all vaccine types (although there is a trend towards mRNA vaccines), and the presence of autoimmune features, such as the detection of autoantibodies and IgG elevation [28,29,30]. The most common symptoms are fatigue and jaundice, with a mixed type of hepatocellular injury (which is not typical for idiopathic AIH). The histological findings are consistent with AIH, although findings of DILI, such as eosinophilic infiltrates, relatively low plasmacytic infiltrates, and the predominance of T- over B-cells, are not rare. Most patients respond well to immunosuppression and remissions are apparently durable. 

Besides the good and prompt response to treatment, there have been reports of deaths because of liver failure [54,57,65,81]. Those cases should alert physicians to the importance of thorough monitoring of selected patients who may be at a higher risk. For instance, AIH-related or other autoantibodies may suggest overt autoimmunity. Moreover, a genetic predisposition may be a factor. In idiopathic AIH, there is a clear association between the expression of the class II HLA phenotype and the DRB1 gene locus. HLA phenotyping was performed in five cases of patients with AIH following SARS-CoV-2 vaccination, and the gene loci detected included A1, B8, DRB1, DQA1, DQB1, DR3, and DR4. In addition, a detailed examination of medical history should be carried out. Screening for liver diseases or medications that could potentiate a vaccine-induced liver injury, and for personal or family history of ADs, is essential. Importantly, patients with pre-existing AIH should be closely monitored. The expert panel consensus by the American Association for the Study of Liver Disease (AASLD) for COVID-19 vaccination on patients with chronic liver disease has not issued any special notice for patients with AIH, other than the recommendation for vaccination [104]. With the increasing reports of post-SARS-CoV-2 vaccination AIH flares, those patients may be at a higher risk than the general population, and we warrant for vigilance. Moreover, the aggressive use of corticosteroids is suggested for patients that are clinically deteriorating, while more evidence is needed for the role of other immunosuppressants used in idiopathic AIH (e.g., azathioprine) or various immunotherapies used in COVID-19 treatment [105].

Finally, based on our experience and on the provided literature evidence, we would suggest LFTs be performed 10–14 days after vaccination on individuals with pre-existing AIH, liver disease, a liver injury after previous vaccination, positive AIH-related autoantibodies or abnormal LFTs prior to vaccination, and known HLA phenotypes associated with AIH. Patients without relevant medical history and mild elevation of liver enzymes may be monitored until remission. For all other cases, we suggest hepatologist counselling for corticosteroid introduction and close follow up. There is not sufficient evidence for the time until symptom resolution, so we would advise for close monitoring until complete liver enzyme normalization and/or treatment withdrawal. Future vaccination plans for patients with severe IM-DILI are challenging. It is clear, though, that the benefits of vaccination significantly outweigh the risks. In fact, comparing infection- versus vaccine-induced autoimmune reactions, the latter generally show a lower incidence, with most of them displaying a milder and self-limiting clinical course [40]. Thus, such reactions should not exclude patients from vaccination and decisions should be individualized. We believe that the rotation of vaccine types, along with close monitoring, based on the physician’s discretion is reasonable.

## 8. Conclusions

Emerging reports suggest that hepatitis with varying degrees of autoimmune features may be associated with vaccination for SARS-CoV-2. As the condition is rare and correlation is difficult, physicians should be alert to such cases. Although the condition resolves in most cases, more severe presentations should be managed promptly with corticosteroids. The close monitoring of selected patients, including those with pre-existing AIH, is reasonable for at least 10 days after vaccination. More evidence is needed for better characterization of this clinical entity.

## Figures and Tables

**Figure 1 vaccines-10-02073-f001:**
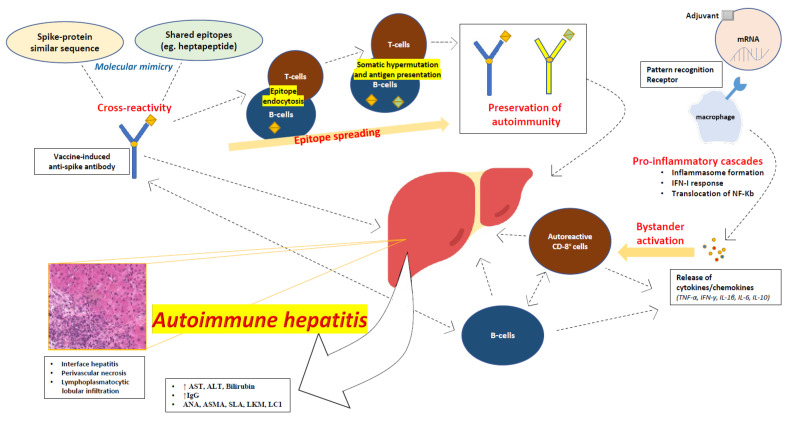
Proposed mechanisms of autoimmune hepatitis following SARS-CoV-2 vaccination.

**Table 1 vaccines-10-02073-t001:** Reported cases and main characteristics of autoimmune hepatitis following SARS-CoV-2 vaccination.

Case	Publication	Gender, Age	Type of Vaccine	Number of Doses	History of Liver Disease or AD	Symptoms, Signs, Laboratory Results/Outcome	Latency (Days)	Serology	Increase in IgG	Biopsy Consistent with AIH
1	[52,53] *	4× F/ 1× M, 62^+^	3× mRNA, 2× Viral vector	n/a	One patient had a probable diagnosis of AIH	All 5 cases demonstrated LFT abnormalities. Resolution in all 5 patients. Four of them needed immunomodulators (corticosteroids–azathioprine) and one did not need any drug remedy.	19^+^	All 5 cases were ANA positive	All 5 cases demonstrated elevation of IgG	Four patients had liver biopsies consistent with AIH, one patient did not have liver biopsy
2										
3										
4										
5										
6	[54]	M, 35	Viral vector	1	No	Fatigue, fever, headache, vomiting, abdominal pain, jaundice, ↑ ATs, ↑ Bil, liver failure, DIC. Death due to fulminant hepatitis, despite high-dose corticosteroids.	5	Negative	n/a	Not consistent with AIH
7	[55]	F, 52	Inactivated	2	No	Fatigue, jaundice, ↑ ATs, ↑ Bil. Rapid improvement with corticosteroids–azathioprine.	7	ANA, ASMA	Yes	Yes
8	[56]	F, 35	mRNA	2	AIH	↑ ATs. Flare of AIH. Remission without medical intervention.	14	ASMA	n/a	Biopsy confirmed AIH 5 years ago
9	[57]	F, 59	Viral vector	2	No	Jaundice, ↑ ATs, ↑ Bil. Responded well to high-dose corticosteroids.	12	Negative	Yes	Yes
10		F, 63	Viral vector	1	PSC	Jaundice, ↑ ATs, ↑ Bil. Initially improved with corticosteroids, but soon relapsed and died due to sepsis.	14	ANA	Yes	Yes
11		F, 72	2× Inactivated, booster with mRNA	3	No	Jaundice, ↑ ATs, ↑ Bil. Responded well to high-dose corticosteroids.	10	AMA	Yes	Yes
12	[58]	F, 82	mRNA	1	Treated HCV	Anorexia, fatigue, ↑ ATs, ↑ Bil. Resolution with corticosteroids.	4	ANA	Yes	Yes
13	[59]	F, 52	mRNA	1	ICIs for lung cancer	Diarrhea (ICI-related colitis), ↑ ATs, ↑ Bil. Resolution with corticosteroids.	10	Negative	No	Biopsy not compatible with AIH
14	[60]	F, 77	mRNA	2	No	Disorientation, vomiting, nausea, jaundice, ↑ ATs, ↑ Bil. HLA-DR4. Resolution with corticosteroids–azathioprine.	2	ANA, AMA	No	Yes
15		M, 23	mRNA	2	No	Mononucleosis-like syndrome, jaundice, ↑ ATs, ↑ Bil. HLA-DR3. Resolution with corticosteroids.	10	Negative	No	Yes
16	[48]	M, 52	mRNA	Episodes after both 1st and 2nd dose.	No	Nausea, fatigue, anorexia, pruritus, jaundice, ↑ ATs, ↑ Bil. Peripheral anti-spike CD8 T-cells. Resolution without medical treatment. Relapse after 2nd dose. Resolution with corticosteroids.	First episode 10 days, 2nd episode 41 days after first dose	ANA, ASMA, AMA	Yes	Yes, panlobular CD8 T-cell infiltration
17	[61]	F, 27	mRNA	2	No	Nausea, vomiting, headache, fever, jaundice, ↑ ATs, ↑ Bil. Resolution with corticosteroids.	6	ANA	Yes	Yes, along with eosinophilic infiltrates
18	[62]	M, 62	Inactivated	2	No	Weakness, jaundice, pruritus, weight loss, ↑ ATs, ↑ Bil. Resolution without corticosteroids.	3	Negative	n/a	Yes, along with eosinophilic infiltrates
19	[63]	F, 63	mRNA	2	No	Abdominal pain, nausea, jaundice, dark urine, acholic stools, ↑ ATs, ↑ Bil. Resolution with corticosteroids–azathioprine.	53	Negative	Yes	Yes
20	[64]	M, 79	Viral vector	Symptoms started after the 1st dose; however, he received a 2nd dose before hospital evaluation.	No	Abdominal pain, jaundice, pruritus, dark urine, acholic stools, ↑ ATs, ↑ Bil. Resolution with corticosteroids–azathioprine.	15	ANA, ASMA	Yes	Yes, along with eosinophilic infiltrates
21	[65]	F, 80	mRNA	2	No	Similar clinical manifestations for all three patients; fatigue, pruritus, jaundice, ↑ ATs, ↑ Bil. Two of them responded well to corticosteroids, one developed liver failure despite treatment and died.	10	All patients were ANA positive; one of them was ASMA positive	All 3 patients demonstrated IgG elevation	All 3 patients had liver biopsy consistent with AIH
22		F, 73	mRNA	1			21			
23		F, 68	Viral vector	1			20			
24	[66]	F, 80	mRNA	2	No	Jaundice, ↑ ATs, ↑ Bil. Improvement with corticosteroids.	10	All patients were ANA positive; one was AMA positive	All 3 patients demonstrated IgG elevation	All 3 patients had liver biopsy consistent with AIH
25		F, 75	mRNA	2	No	↑ ATs, ↑ Bil. Improvement with corticosteroids.	4			
26		F, 78	mRNA	1	PBC	Fever, malaise, ↑ ATs, ↑ Bil. Improvement with corticosteroids.	7			
27	[67]	F, 57	Inactivated	She developed symptoms after the 1st dose, but she did not seek medical advice and received a 2nd dose.	No	Jaundice, dark urine, acholic stools, pruritus, ↑ ATs, ↑ Bil. Resolution with corticosteroids–azathioprine.	14	ANA, SSA	Yes	Yes
28	[68]	F, 61	mRNA	n/a	Hashimoto thyroiditis	Malaise, fatigue, anorexia, nausea, jaundice, ↑ ATs, ↑ Bil. Resolution with corticosteroids–azathioprine.	14	ANA, ASMA	Yes	Yes
29	[69]	F, 65	mRNA	1	No	Abdominal pain after vaccine injection, jaundice, dark urine, ↑ ATs, ↑ Bil. Resolution with corticosteroids.	14	ANA	Yes	Yes
30	[70]	F, 40	mRNA	2	Sarcoidosis	↑ ATs. Resolution with corticosteroids.	30	ANA	Yes	Yes
31	[71]	F, 36	mRNA	1	PSC, ulcerative colitis	↑ ATs, ↑ Bil. Resolution with corticosteroids.	12	ANA	n/a	Yes, along with eosinophilic infiltrates
32	[72]	M, 63	mRNA	1	No	Fatigue, jaundice, anorexia, ↑ ATs, ↑ Bil. HLA-DRB1, DQA1, DQB1. Resolution with corticosteroids.	7	ANA, anti-gastric parietal	Yes	Yes
33	[73]	F, 71	mRNA	1	No	Jaundice, ↑ ATs, ↑ Bil. Resolution with corticosteroids.	4	ASMA	Yes	Yes, along with eosinophilic infiltrates
34	[74]	M, 36	Viral vector	1	No	Pruritus, ↑ ATs, ↑ Bil. Resolution with corticosteroids.	26	ANA	No	Yes
35	[75]	F, 56	mRNA	1		Fatigue, jaundice, anorexia, ↑ ATs, ↑ Bil. Resolution with corticosteroids.	40	ANA, ASMA	Yes	Yes, along with eosinophilic infiltrates
36	[76]	F, 80	mRNA	2	Hashimoto thyroiditis	Jaundice, dark urine, ↑ ATs, ↑ Bil. Improvement with corticosteroids.	7	ANA	Yes	Yes
37	[77]	F, 35	mRNA	1	No	Jaundice, pruritus, dark urine, ↑ ATs, ↑ Bil. Improvement with corticosteroids.	6	ANA, anti-ds DNA	No	Yes, along with eosinophilic infiltrates
38	[78]	F, 52	mRNA	1	No	Nausea, jaundice, ↑ ATs, ↑ Bil. Resolution with corticosteroids.	14	ANA, ASMA	Yes	Yes, along with eosinophilic infiltrates
39	[79]	F, 43	mRNA	First symptoms presented after 1st dose; however, she received a 2nd vaccine dose.	No	Jaundice, pruritus, ↑ ATs, ↑ Bil. Resolution with corticosteroids.	15	Negative	No	Yes
40	[80]	F, 41	mRNA	First symptoms presented after 1st dose and lasted for 3 weeks; however, she received a 2nd vaccine dose, which flared the symptoms.	No	Abdominal pain, nausea, vomiting, jaundice, ↑ ATs, ↑ Bil. Resolution with corticosteroids.	1	ANA, ASMA, SLA, anti-liver cytosol	Yes	Yes
41	[81]	F, 38	Viral vector	1	No	Jaundice, fatigue, pedal edema, dark urine, ↑ ATs, ↑ Bil. Resolution with corticosteroids.	7	ANA	Yes	Yes
42		M, 62	Viral vector	1	History of jaundice episodes (unknown etiology) with spontaneous resolution	Fever, anorexia, jaundice, ↑ ATs, ↑ Bil. Patient was managed with corticosteroids and plasmapheresis with poor response and died 3 weeks later.	13	Negative	n/a	Compatible, but not typical
43	[82]	M, 47	mRNA	First symptoms presented after 1st dose; however, he received a 2nd vaccine dose, which flared symptoms.	No	Jaundice, malaise, ↑ ATs, ↑ Bil. Resolution with corticosteroids.	3	ANA	Yes	Yes, along with eosinophilic infiltrates
44	[83]	M, 76	mRNA	1	Hashimoto thyroiditis	Dark urine, weight loss, fatigue, ↑ ATs, ↑ Bil. Resolution with corticosteroids–azathioprine.	3	ANA, ASMA, anti-actin, anti-neutrophilic cytoplasmic	Yes	Yes
45	[84]	F, ≈30	Viral vector	1	No	Jaundice, dark urine, acholic stools, fatigue, ↑ ATs, ↑ Bil. Resolution with corticosteroids.	10	ANA	Yes	Yes, along with findings compatible with drug toxicity
46	[85]	F, 30	mRNA	2	Hashimoto thyroiditis, family history of ADs	Fatigue, anorexia, arthralgia, dark urine, ↑ ATs, ↑ Bil. Spontaneous resolution.	≈30	ANA	Yes	Yes
47		M, 26	mRNA	2	Vitiligo	↑ ATs, ↑ Bil. HLA-A1, B8, DR3. Persistence of LFT abnormalities, as declined treatment with corticosteroids.	≈30	ANA	Yes	Yes
48		F, 21	mRNA	↑ ATs before vaccination. Workup was consistent with AIH diagnosis. Flare of symptoms after both 1st and 2nd vaccine dose.	AIH	↑ ATs. HLA-A1, B8, DR3. First episode remitted spontaneously. After 2nd dose, resolution with corticosteroids.	n/a	ANA, ASMA	No	Yes
49	[86]	M, 76	mRNA	n/a	Liver cirrhosis secondary to PBC	↑ ATs, ↑ Bil. Resolution with corticosteroids–azathioprine.	≈30	ANA	Yes	Yes
50	[87]	F, 32	mRNA	3	Liver transplantation due to AIH	↑ ATs, ↑ Bil. Remission resolution with corticosteroids–azathioprine.	21	Liver cytosol antibody 1	No	Yes
51	[88]	F, 57	mRNA	1	No	Fatigue, ↑ ATs, ↑ Bil, ↑γGT, ↑ALP. Resolution without administration of immunomodulation (patient treated with ursodeoxycholic acid).	14	ANA, AMA	No	Yes, along with findings of granulomatous, non-suppurative cholangitis with destruction and proliferation of bile duct (AIH/PBC variant)

AD: autoimmune disease; AIH: autoimmune hepatitis; AMA: antimitochondrial antibodies; ANA: antinuclear antibodies; ASMA: anti-smooth muscle antibody; AT: aminotransferases; Bil: bilirubin; DIC: disseminated intravascular coagulation; F: female; ICI: immune checkpoint inhibitors; LFT: liver function test; n/a: not available; M: male; PBC: primary biliary cholangitis; PSC: primary sclerosing cholangitis; SSA: Sjögren syndrome antigen A. * Torrente et al. case overlaps with the case series reported by Izagirre et al. ^+^ Median. ↑ Increased.

**Table 2 vaccines-10-02073-t002:** Probability for autoimmune hepatitis diagnosis and causality of vaccination.

Cases	Publication	Simplified Criteria for AIH		Naranjo Score for Vaccine Causality	
		Score	Probability	Score	Probability
1	[52] *	7 ^+^	Definite	5	Probable
2	[53]	8 ^+^	Definite	May be 2 to 5 for each patient	Possible/probable
3		8 ^+^	Definite		
4		8 ^+^	Definite		
5		n/a	n/a		
6	[54]	2	Non-probable	2	Possible
7	[55]	8	Definite	7	Probable
8	[56]	4	Non-probable	6	Probable
9	[57]	4	Non-probable	7	Probable
10		6	Probable	7	Probable
11		5	Non-probable	7	Probable
12	[58]	6	Probable	7	Probable
13	[59]	0	Non-probable	3	Possible
14	[60]	6	Probable	7	Probable
15		4	Non-probable	7	Probable
16	[48]	7	Definite	9	Definite
17	[61]	7	Definite	7	Probable
18	[62]	3	Non-probable	6	Probable
19	[63]	5	Non-probable	7	Probable
20	[64]	7	Definite	7	Probable
21	[65]	≥6 for each patient	At least probable AIH for each patient	7	Probable
22				7	Probable
23				6	Probable
24	[66]	6	Probable	7	Probable
25		7	Definite	7	Probable
26		7	Definite	7	Probable
27	[67]	7 ^+^	Definite	7	Probable
28	[68]	7 ^+^	Definite	7	Probable
29	[69]	8 ^+^	Definite	7	Probable
30	[70]	8	Definite	7	Probable
31	[71]	6	Probable	7	Probable
32	[72]	7	Definite	7	Probable
33	[73]	7	Definite	7	Probable
34	[74]	6	Probable	7	Probable
35	[75]	≥7	Definite	7	Probable
36	[76]	8	Definite	7	Probable
37	[77]	6	Probable	7	Probable
38	[78]	7	Definite	7	Probable
39	[79]	4	Non-probable	7	Probable
40	[81]	8 ^+^	Definite	9	Definite
41	[81]	7	Definite	7	Probable
42		3	Non-probable	3	Possible
43	[82]	≥6	At least probable	9	Definite
44	[83]	8 ^+^	Definite	7	Probable
45	[84]	6 ^+^	Probable	4	Possible
46	[85]	7	Definite	6	Probable
47		7	Definite	6	Probable
48		6	Probable	8	Probable
49	[86]	7	Definite	7	Probable
50	[87]	6	Probable	7	Probable
51	[88]	6	Probable	6	Probable

AIH: autoimmune hepatitis; n/a: not available. * Torrente et al. case overlaps with the case series reported by Izagirre et al. ^+^ Provided by the author.

## Data Availability

The data presented in the review are available in the article and Appendix A.

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
