# Peer review of "Vaccine-Related Autoimmune Hepatitis: Emerging Association with SARS-CoV-2 Vaccination or Coincidence?"

_vaccines, 2022, doi:10.3390/vaccines10122073_

Round 1
Reviewer 1 Report
Ioannis Trontzas and colleagues present a high quality and well-written review manuscript focused on vaccine-related autoimmune hepatitis and its emerging association with SARS-CoV-2 vaccination.
Authors performed a literature review to identify articles regarding the association of AIH with vaccination, emphasizing on SARS-CoV-2 vaccines, and the proposed mechanisms. Authors then performed a literature search for AIH-like cases following SARS-CoV-2 vaccination and they evaluated the included cases for AIH diagnosis, using Simplified Diagnostic Criteria (SDC), and for vaccination causality, using Naranjo score for adverse drug reactions.
Authors identified 51 AIH-like cases following SARS-CoV-2 vaccination. Forty cases (80%) were characterized as “probable”, “at least probable” or “definite” for AIH diagnosis according to SDC. Forty cases (78.4%) were characterized as “probable”, four (7.8%) as “possible”, and three (5.8%) as “definite” for vaccine related AIH according to Naranjo score.
Finally, authors conclude that SARS-CoV-2 vaccine-related AIH carries several phenotypes and, although most cases resolve, immunosuppressive therapy seems to be necessary. Early diagnosis is mandatory and should be considered in any patient with acute or chronic hepatitis after SARS-CoV-2 vaccination, especially in those with pre-existing liver disease.
Overall, the manuscript is highly valuable for the scientific community and should be accepted for publication.
======================
Other comments to authors:
1) Please check for typos throughout the manuscript.
2) Authors are kindly encouraged to cite the following article that describes the use of certain immunotherapies against COVID-19.
DOI: 10.3390/biomedicines9010059
Reviewer 2 Report
It is a review paper discussing the association of SARS-CoV-2 vaccination and vaccine-related autoimmune hepatitis. It is a well-written review paper summarized reasonable among of related publications into two nice tables, I only have few minor suggestions.
1. Why case 1 to 5 are summarized in one row for Table 1. Unless with special reason, it shall be separated case 1 and case 2 to 5 as shown in Table 2.
2. For the columns for authors in Table 1 and 2, name of first authors are not necessary. Suggest only show the number of the reference.
3. Few words or phrases are bolded in Table 1 and 2. Without any special reason, please un-bolded them. Otherwise, please provide reason and put it as footnote under the table why they are bolded.
4. In terms of the liver injury cases resembling autoimmune hepatitis (AIH) following SARS-CoV-2 vaccination, it shall be very low (based on the data provided by the authors). It is better to avoid using the ‘substantial’ in the first sentence of abstract.
